# Robust Control Based on Modeling Error Compensation of Microalgae Anaerobic Digestion

**Mariana Rodríguez-Jara** [1], **Alejandra Velasco-Pérez** [2,*], **Jose Vian** [3], **Sergio E. Vigueras-Carmona** [4] **and Héctor Puebla** [1]

1 Universidad Autónoma Metropolitana Azcapotzalco, Ciudad de México 02128, Mexico
2 Universidad Veracruzana Campus Orizaba, Orizaba Veracruz 94300, Mexico
3 Universidad Politécnica de Huatusco, Veracruz 94116, Mexico
4 Tecnológico de Estudios Superiores de Ecatepec, Estado de México 55210, Mexico
* Correspondence: alvelasco@uv.mx; Tel.: +52-555318-9000

**Abstract:** Microalgae are used to produce renewable biofuels (biodiesel, bioethanol, biogas, and biohydrogen) and high-value-added products, as well as in bioremediation and $CO_2$ sequestration tasks. In the case of anaerobic digestion of microalgae, biogas can be produced from mainly proteins and carbohydrates. Anaerobic digestion is a complex process that involves several stages and is susceptible to operational instability due to various factors. Robust controllers with simple structure and design are necessary for practical implementation purposes and to achieve a proper process operation despite process variabilities, uncertainties, and complex interactions. This paper presents the application of a control design based on the modeling error compensation technique for the anaerobic digestion of microalgae. The control design departs from a low-order input–output model by enhancement with uncertainty estimation. The results show that achieving desired organic pollution levels and methanogenic biomass concentrations as well as minimizing the effect of external perturbations on a benchmark case study of the anaerobic digestion of microalgae is possible with the proposed control design.

**Keywords:** microalgae; anaerobic digestion; process control; robust control





## 1. Introduction

The world's rising energy demand, the depletion of fossil fuel resources, and environmental concerns about climate change are major drivers to research biofuels [1,2]. Indeed, biofuels could be a long-term replacement for fossil fuels. Microalgae produce renewable and sustainable biofuels (biodiesel, bioethanol, biogas, and biohydrogen), pharmacological products, and food ingredients [3–5]. Microalgae has several advantages against other renewable biomass sources. For instance, fast growth rate, high oil yield, the use of non-arable land for algae cultivation, growth in various water sources, and carbon dioxide ($CO_2$) mitigation [6].

Microalgae can be used to produce usable energy via multiple routes. Biodiesel can be produced through transesterification between algal oil (the triacylglycerol (TAG)) and alcohol with a catalyst [7]. Bioethanol and biogas can be made from the saccharides in microalgal biomass via fermentation or anaerobic digestion (AD) [5,8,9]. The starch degradation from microalgae can also be used to produce biohydrogen [5,10,11]. Furthermore, the residue is considered waste after extracting the lipid for biodiesel production and other components for cosmetic and pharmaceutical purposes. The valorization of this microalgae residue can also be used for biogas production via AD [12]. Thus, the digestion of microalgae to provide an additional source of biofuel is an attractive possibility. Indeed, recent advances in AD for biogas production using as a feedstock both microalgae and macroalgae have been reviewed and described by several authors [13–16].

AD has a high capacity to degrade concentrated and difficult substrates (plant residues, animal wastes, food industry wastewater, and so forth), produces very little sludge, requires little energy, and, in some cases, it can even recover energy using methane combustion [17,18]. Challenges in the optimization and control of AD include limited process knowledge, nonlinearities, unmodeled dynamics, unknown internal and external noises, environmental influences, and time-varying parameters [19]. Another issue is the feed of the digester, which is primarily waste; consequently, a constant and nonpolluted inflow in the digester cannot always be guaranteed [20].

The coupling of AD and microalgae cultivation has been proposed as a viable option to produce renewable biofuel [21,22]. This process not only recovers the energy stored in the microalgae biomass but also leads to ammonium and phosphate release, which can be a source of nutrients for the microalgae culture [16]. Furthermore, the generation of $CO_2$ in AD can be used to grow photoheterotrophic microalgae populations in conjunction with either artificial or solar light as a source of energy [23].

Microalgae AD's main process objectives are reducing the organic pollution level and biogas production [24,25]. After lipid extraction for biodiesel production and other valuable products such as amino acids, the microalgae residue contains cellulose, lignin, lipid, and protein. In the first case, the microalgae residue requires appropriate treatment for its final disposal. In the second case, the organic residue can be used as a resource for biogas production to enhance the biofuel production obtained from the microalgae.

Nevertheless, due to their inherent complexity, the control and optimization of such coupled microalgae cultivation–anaerobic digester systems present many challenges [22]. For instance, operational issues of the AD of microalgae include possible process instability caused by the inhibition effects of inorganic nitrogen and volatile fatty acids, which can give rise to the disappearance of the methanogenic bacteria or even the so-called washout state of the process [25–27]. Hence, the complexity and operational instability of the AD due to variations in the process operating conditions have probably inhibited the industrial AD processing of plant residues, including macro and microalgae residues.

Thus, control designs are required to achieve the main objectives of microalgae AD. To the author's knowledge, the control design of microalgae AD has been scarcely addressed in the literature. One aspect motivating the design of control schemes for AD of microalgae is the development of valuable and reliable mathematical models of the primary process variables [28–30]. In particular, Mairet et al. [28] introduced a model for AD of microalgae (named the "MAD model"), which was validated on experimental data. The proposed model has allowed the derivation of ideal values for operating parameters, studied these parameters' influence on the process's qualitative behavior, and designed a model-based controller.

This paper addresses the robust control design of AD of microalgae using a simple, versatile and practical approach via modeling error compensation (MEC) ideas [31,32]. Two control problems in the AD of microalgae are addressed: (i) regulation of the organic pollution level and (ii) regulation of the methanogenic biomass concentration. Although the control literature on AD is vast, to the best of the author's knowledge, the proposed controller has not previously been used for microalgae AD in the addressed control problems. Furthermore, it has been pointed out that it is necessary to develop and apply reliable, practical, and robust controllers in bioprocesses and advanced control strategies to renewable fuel production and $CO_2$ sequestration applications [33–36]. Numerical simulations show that the proposed controller may achieve robust regulation of the organic pollution level and the methanogenic biomass concentration. Thus, our paper's main contribution is the introduction of a robust and practical controller for the continuous AD of microalgae that achieves good closed-loop performance despite model uncertainties and external perturbations.

This work is organized as follows: In Section 2, the benchmark MAD model is described, as well as the main ideas of the robust controller. Section 3 presents the closed-loop performance of the proposed controller for two control tasks: (i) regulation of the organic

pollution level and (ii) regulation of the methanogenic biomass concentration. In Section 4, the discussion and practical implications of the results are presented. Finally, in Section 5, some concluding remarks are presented.

## 2. Materials and Methods

This section first presents the benchmark MAD model and the general methodology of the proposed controller based on MEC ideas.

### 2.1. AD of Microalgae

The AD process is performed by a community of four groups of bacteria (hydrolytic, fermentative, acetogenic, and methanogenic) that decompose the organic matter in four steps: hydrolysis, acidogenesis, acetogenesis, and methanogenesis [17,18]. Hydrolysis of polymeric compounds such as carbohydrates, proteins, and fats is the first step in anaerobic digestion. Fermentative and acetogenic bacteria metabolize simpler organic molecules produced via hydrolysis to hydrogen, acetate, formate, and carbon dioxide, which are transformed into methane by the methanogenic bacteria. In particular, acetate is a key intermediate contributing to a significant part of the produced methane [37]. In microalgae AD, proteins are hydrolyzed to amino acids by extracellular enzymes. Then, amino acid fermentation is performed by anaerobic and facultative anaerobic bacteria. Methanogenesis converts the hydrogen and acetic acid to methane gas and carbon dioxide via archaea, which are more sensitive to toxic compounds and exhibit lower growth rates [38].

Since the seminal study of Golueke et al. [27], several authors have considered the anaerobic digestion of algal biomass, which were reviewed by Ward et al. [16], Uggetti et al. [9], and Milledge et al. [25]. These studies pointed out some features in the anaerobic digestion of microalgae: (i) difficult digestibility, mainly due to the cell walls composition of some microalgae; (ii) low carbon-to-nitrogen (C/N) ratios associated with high nitrogen content in the microalgae; and (iii) the protein degradation of the biomass microalgae results in the formation of ammonia, which can be inhibitory.

Indeed, when protein-rich microalgae are subjected to AD, the bioprocess can be affected mainly by the hydrolysis and methanogenesis steps due to the high amount of nitrogen released in the form of ammonium. Furthermore, some features of microalgae cultivation, including its intracellular and cell wall composition, lead to strain-specific microalgae AD efficiency [39].

The above operational issues have motivated different studies to improve the technology and operational performance of AD of microalgae. These studies include chemical and thermal pretreatment of algal biomass [16], coupling AD with microalgae cultivation systems [21], co-digestion with substrates with a high C/N [40], and modeling approaches to improve the process prediction, set optimal conditions, and develop and apply different control designs [28,41–43].

### 2.2. MAD Model

Based on principal component analysis (PCA), as well as the available liquid-phase measurements (total COD, inorganic nitrogen, and VFA concentrations) and the methane flow rate measurements, Mairet et al. [28] set a three biochemical reaction scheme to derive a model of the microalgae anaerobic digestion. The proposed model was developed and calibrated for the AD of *Chlorella vulgaris*, a microalgae species commonly used for biofuel production. Experimental arrangement and conditions were characterized and described by Mairet et al. [28].

The main model assumptions are [28]: (i) The microalgae biomass is divided into three substrates, $S_1$ (sugar and lipids excluding nitrogen), $S_2$ (proteins), and $S_I$ (inert substrate). (ii) $S_1$ and $S_2$ are degraded to VFAs ($S_3$) via hydrolysis–acidogenesis–acetogenesis by the bacterial populations $X_1$ and $X_2$, respectively. (iii) $S_3$ is converted to methane by methanogenic population $X_3$. (iv) The specific growth rates for the hydrolysis–acidogenesis–acetogenesis reactions are modeled as Contois functions. (v) The methanogenesis-specific

growth rate is modeled with a Haldane function with a multiplicative ammonia inhibition term. (vi) pH values for the AD of *C. vulgaris* are in the range of 6.0 < pH < 7.5. (vii) The total inorganic carbon concentration (C) is the sum of the dissolved carbon dioxide concentration $CO_2$ and the bicarbonate concentration $HCO_3$. (viii) The total inorganic nitrogen (N) is the sum of free ammonium and ammonium ions. Ammonium ions are consumed in the hydrolysis–acidogenesis–acetogenesis of $S_1$ and the methanogenesis of VFAs. (ix) The anaerobic digester is a 1-L ($V_{liq}$) continuous perfectly stirred reactor with 0.1-L headspace ($V_{gas}$). (x) The microalgae fed is composed by fractions of sugars-lipids $\beta_1$, proteins $\beta_2$, and inerts $\beta_I$. (xi) The operation temperature ($T_{op}$) is maintained constant. (xii) All of the produced methane is transferred to the headspace.

The mathematical model is calculated as follows [28]:

$$\frac{dS_1}{dt} = D(\beta_1 S_{in} - S_1) - \alpha_1 \mu_1 X_1$$
$$\frac{dS_2}{dt} = D(\beta_2 S_{in} - S_2) - \alpha_5 \mu_2 X_2$$
$$\frac{dS_I}{dt} = D(\beta_I S_{in} - S_I)$$
$$\frac{dS_3}{dt} = -DS_3 + \alpha_3 \mu_1 X_1 + \alpha_6 \mu_2 X_2 - \alpha_9 \mu_3 X_3$$
$$\frac{dX_1}{dt} = -DX_1 + \mu_1 X_1$$
$$\frac{dX_2}{dt} = -DX_2 + \mu_2 X_2$$
$$\frac{dX_3}{dt} = -DX_3 + \mu_3 X_3$$
$$\frac{dN}{dt} = D(N_{in} - N) - \alpha_2 \mu_1 X_1 + \alpha_7 \mu_2 X_2 - \alpha_{10} \mu_3 X_3$$
$$\frac{dC}{dt} = D(C_{in} - C) + \alpha_4 \mu_1 X_1 + \alpha_8 \mu_2 X_2 + \alpha_{12} \mu_3 X_3 - \rho_{CO_2}$$
$$\frac{dP_{CO_2}}{dt} = -\frac{q_{gas}}{V_{gas}} P_{CO_2} + \frac{V_{liq} R T_{op}}{V_{gas}} \rho_{CO_2}$$
$$\frac{dP_{CH_4}}{dt} = -\frac{q_{gas}}{V_{gas}} P_{CH_4} + \frac{V_{liq} R T_{op}}{V_{gas}} \rho_{CH_4}$$
$$\frac{dz}{dt} = D(z_{in} - z)$$

With

$$\mu_1 = \mu_{1,max} \frac{S_1}{k_{s1} X_1 + S_1}$$
$$\mu_2 = \mu_{2,max} \frac{S_2}{k_{s2} X_2 + S_2}$$
$$\mu_3 = \mu_{3,max} \frac{S_3}{k_{s3} + S_3 + \frac{S_3^2}{k_I}} \frac{K_{I_{NH3}}}{K_{I_{NH3}} + NH_3}$$
$$NH_3 = \frac{K_N}{K_N + h} N, \ h = 10^{-pH}$$
$$\rho_{CO_2} = k_L a \left( \frac{h}{k_C + h} C - K_{H,CO_2} P_{CO_2} \right)$$
$$\rho_{CH_4} = \alpha_{11} \mu_3 X_3$$

where $D$ is the dilution rate, $\alpha_i$ are the stoichiometric parameters ($i = 1,..., 12$). $S_{in}$, $N_{in}$, $C_{in}$, and $z_{in}$ are the input concentrations of organic matter, inorganic nitrogen, inorganic carbon, and alkalinity, respectively. $P_{CO_2}$ and $P_{CH_4}$ are the partial pressures of $CO_2$ and $CH_4$, and $\rho_{CO2}$ and $\rho_{CH4}$ are their liquid–gas transfer rates.

The biogas flow rate is calculated by

$$q_{gas} = \max \left( 0, k_v \left( P_{CO_2} + P_{CH_4} - P_{atm} \right) \right)$$

With $k_v$ as the pipe resistance coefficient and $P$ as the atmospheric pressure.

For control design purposes, the state vector is shown as $x = [x_1, x_2, x_3, x_4, x_5, x_6, x_7, x_8, x_9, x_{10}, x_{11}, x_{12}]^T = [S_1, S_2, S_I, S_3, X_1, X_2, X_3, N, C, P_{CO2}, P_{CH4}, z]^T$. The control input is the dilution rate, i.e., $u = D$. The nominal operation is simulated with the following parameter values [28]: $S_{in} = 29.5$ gCOD/L, $\beta_1 = 0.3$, $\beta_2 = 0.4$, $\beta_I = 0.3$, $\mu_{1,max} = 0.3$ d$^{-1}$, ks$_1 = 2.11$ gCOD/L, $\mu_{2,max} = 0.053$ d$^{-1}$, ks$_2 = 0.056$ gCOD/L, $\mu_{3,max} = 0.14$ d$^{-1}$, ks$_3 = 0.02$ gCOD/L, k$_{I3} = 16.4$ gCOD/L, k$_N = 1.1 \times 10^{-9}$, k$_{INH3} = 1.1 \times 10^{-9}$,

$\alpha_1 = 12.5$, $\alpha_2 = 0.0062$, $\alpha_3 = 11.5$, $\alpha_4 = 0.03$, $\alpha_5 = 9.1$, $\alpha_6 = 8.1$, $\alpha_7 = 0.054$, $\alpha_8 = 0.03$, $\alpha_9 = 20$, $\alpha_{10} = 0.062$, $\alpha_{11} = 0.3$, $\alpha_{12} = 0.2$, $k_L a = 5 \text{ d}^{-1}$, $k_v = 5\text{e}4 \text{ L/d bar}$, $R = 8.314 \times 10^{-2} \text{ bar/M K}$, $T_{op} = 308.15 \text{ K}$, $pH = 7$, $P_{atm} = 1.01325 \text{ bar}$, $V_{liq} = 1 \text{ L}$, $V_{gas} = 0.1 \text{ L}$, $N_{in} = 0.011 \text{ M}$, $C_{in} = 0.019$, $z_{in} = 0.017$.

The steady-state shown by $[x_1, x_2, x_3, x_4, x_5, x_6, x_7, x_8, x_9, x_{10}, x_{11}, x_{12}]^* = [\ 0.289, 1.097,$ $8.85, 0.065, 0.6848, 1.176, 0.8668, 0.0648, 0.0748, 0.405, 0.608, 0.017]$ is obtained with the above parameter values, and a dilution value base of $u = 0.05 \text{ d}^{-1}$. It can be noted that the nominal biomass productivity (i.e., $u \cdot x_6$) is around $0.0433 \text{ gCOD L}^{-1} \text{ d}^{-1}$. Figure 1 shows the effect of a step change on the dilution rate at $t = 500$ d. It is noted that a smooth response is obtained for both methanogenic biomass and the organic pollution level, defined as the sum of the microalgae organic components $S_1$ and $S_2$ and the $S_3$ produced in the microalgae AD. On the other hand, an initial inverse response is observed for both the VFA concentration and the biogas flow.

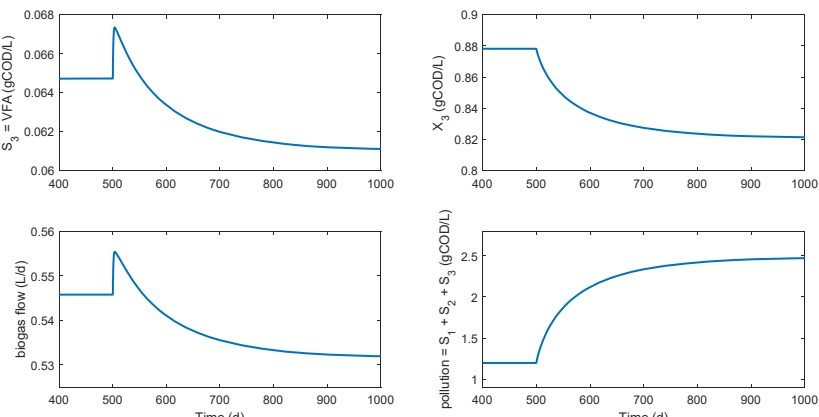

**Figure 1.** Base numerical simulations of two main MAD model variables (VFA and $X_3$ concentrations in gCOD/L), biogas flow (L/d), and organic pollution level (gCOD/L), including the effect of a step change in the dilution rate at $t = 500$ d.

### 2.3. Robust Control Design Based on MEC

Robust control involves quantifying the uncertainties in a nominal process model and designing a controller that copes with these uncertainties to achieve specified performance over the range of operating conditions [44,45]. An approach to developing practicable robust model-based controllers is the MEC approach. The MEC design is based on linear or nonlinear models with lumped bounded uncertainties, which are estimated and compensated with inverse dynamics controllers. The MEC approach was first introduced by Sun et al. [46]. Alvarez-Ramirez [31] extends the original ideas to nonlinear feedback linearizable systems. Rodriguez-Jara et al. [32] present a more straightforward MEC approach to derive practical, robust controllers departing from input–output low-order transfer functions. The MEC approach has been extensively applied and accepted as an effective technique for robustly controlling uncertain nonlinear systems in complex (bio)-chemical processes [47–49].

The control design departs from an input–output first-order model obtained from step response [32], i.e.,

$$G_p(s) = \frac{Y(s)}{U(s)} = \frac{k_p}{\tau_0 s + 1}$$

where $k_p$ and $\tau_0$ are the steady-state process gain and the process time constant. The corresponding first-order input–output model in the time domain is enhanced with lumped model uncertainties $\eta(t)$, including structural uncertainties with bounded variation due to the model reduction $\xi(y(t))$, and constant or persistent external perturbations $\pi(t)$, i.e.,

$$\frac{dy(t)}{dt} = -\frac{1}{\tau_0}y(t) + \frac{k_p}{\tau_0}u(t) + \eta(t)$$

The lumped uncertain term is estimated with a reduced-order observer [31,32],

$$\frac{d\widetilde{\eta}(t)}{dt} = \frac{1}{\tau_e}(\eta(t) - \widetilde{\eta}(t))$$

where $\widetilde{\eta}(t)$ is the estimated modeling error term, and $\tau_e$ is an observer parameter, denoted as the estimation time constant, that modulates the convergence rate of the estimation of the real uncertain term. After algebraic manipulations, the reduced observer can be written as

$$\frac{dw(t)}{dt} = \frac{1}{\tau_0}y(t) - \frac{k_p}{\tau_0}u(t) - \frac{1}{\tau_e}(w(t) + y(t))$$
$$w(0) = -y(0)$$
$$\widetilde{\eta}(t) = \frac{1}{\tau_e}(w(t) + y(t))$$

The controller is proposed based on the model inversion to assign an asymptotic first-order closed-loop behavior and the cancelation of the estimated uncertain term $\widetilde{\eta}(t)$,

$$u(t) = \frac{\tau_0}{k_p}\left(\frac{1}{\tau_0}y(t) - \widetilde{\eta}(t) - \frac{1}{\tau_c}e(t) + \frac{dy_{ref}}{dt}\right)$$

where $y_{ref}(t)$ is the desired set-point, $e(t) = y(t) - y_{ref}(t)$, is the regulation or tracking error, and $\tau_c$ is a controller parameter, denoted as the closed-time constant, that modulates the closed-loop convergence rate to the desired set-point. Based on the process time constant, $\tau_0$, tuning of parameters $\tau_c$ and $\tau_e$ follows the rule [31,32,47]: $0 < \tau_e < \tau_c < \tau_0$.

In this paper, the MEC approach is applied to the MAD benchmark model under the following additional assumptions:

**Assumption 1.** *The dilution rate is the control input, i.e., $u = D$.*

**Assumption 2.** *The control input is subjected to a saturation nonlinearity, i.e., $u_{min} \leq u \leq u_{max}$.*

**Assumption 3.** *The controlled variable is available for control design purposes.*

The following comments are in order:

A.  For optimization and control purposes of AD processes, usually in practice, only a relatively limited number of control actions are possible. These are mostly restricted to the input flow rate or the input of a particular substrate in the feed [19,20]. Therefore, this paper selects the dilution rate (directly related to the input flow rate) as the control input variable.

B.  The minimum and maximum control inputs are selected following previous studies on the operational behavior of the MAD model [30]. The maximum input flow rate must be chosen to prevent the washout condition.

## 3. Results

In this section, the above-described robust control approach is applied to the benchmark MAD model for two control problems: (i) The regulation of the organic pollution level and (ii) the regulation of the methanogenic biomass. The performance of the MEC controller is evaluated for a set-point $y_{ref}$ change at $t$ = 1000 d and an external perturbation of +20% on the input substrate feed $S_{in}$ at $t$ = 1500 d.

To set the references for both control problems, screening on the dilution rate was performed between the minimum and maximum values of the decision variable. Figure 2 shows the results. It is noted that both low organic pollution levels and higher methanogenic biomass concentration are achieved at low dilution values, which corresponds to high hydraulic retention times (HRT). Thus, the proposed references are selected to achieve an organic pollution level around or below 2 gCOD/L, and a methanogenic biomass concentration of around 9 gCOD/L.

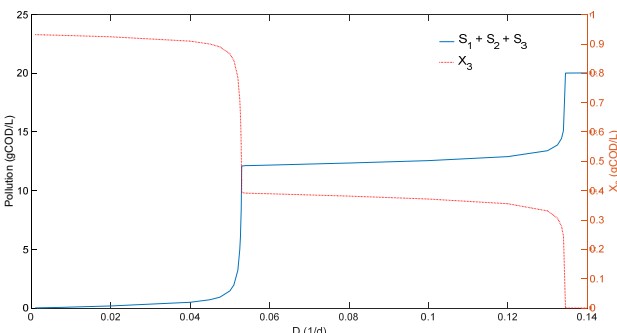

**Figure 2.**  Effect of the dilution on the organic pollution level and methanogenic biomass concentrations.

### 3.1. Control of the Organic Pollution Level

### 3.1.1. Control Problem

The control problem is set as the regulation of the organic pollution level to the desired reference by manipulating the dilution rate. Recently, many laws and regulations have been issued to decrease pollution related to industrial and urban water effluents [50]. In the case of the AD of microalgae, the analogous problem is related to the disposal of the microalgae biomass residues [21,22]. Microalgae residues are generated after extracting valuable products such as amino acids and lipids. Hence, the main goal is to obtain a minimal quantity of output pollutants defined as the sum of the nondegraded organic components in the microalgae $S_1$, $S_2$, and the produced VFAs $S_3$.

### 3.1.2. Numerical Results

The initial step in the design of the MEC controller is to derive the input–output transfer function model between the dilution rate and the organic pollution level. Based on the input–output response shown in Figure 1, the first-order transfer function parameters are provided as $k_p = 515$ and $\tau_0 = 75$.

Figure 3 shows the closed-loop performance of the MEC controller for three sets of controller parameters. The performance of the MEC controller is compared against a conventional PI controller using IMC tuning rules [51].

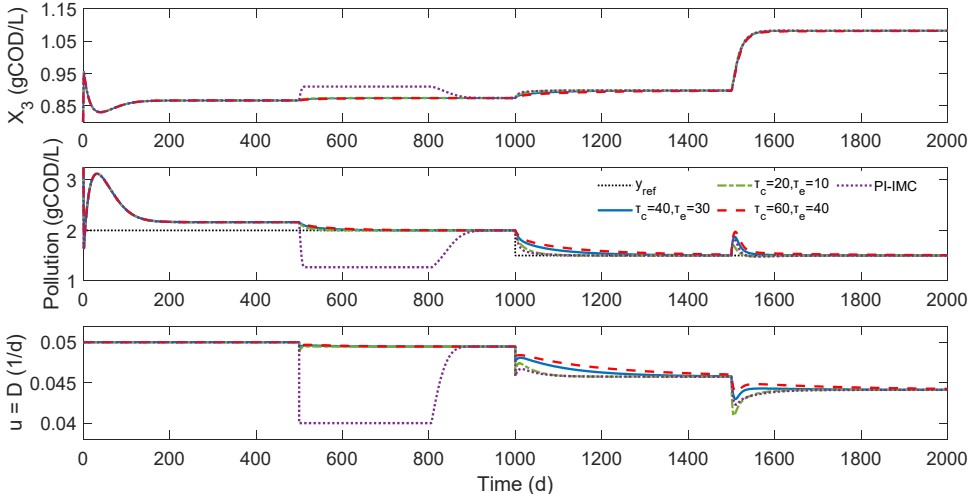

**Figure 3.** Closed-loop performance of the MEC controller for the organic pollution control problem.

It is noted from Figure 3 that the MEC controller can regulate the organic pollution level to the desired references, as well as minimize the effect of the external perturbation. The observed control input adjustment is the following: (i) Initially, the control input is set

at the base value of 0.05 d$^{-1}$, which leads to a steady-state value of 2.16 gCOD/L and a methanogenic biomass concentration of 0.86 gCOD/L. (ii) Once the controller is activated at $t$ = 500 d, to achieve the desired reference of the organic pollution level of 2 gCOD/L, a slight decrease in the dilution rate to 0.0495 d$^{-1}$ is performed to increase the degradation of the organic pollution level via an increase in the HRT. (iii) At $t$ = 1000 d, the dilution rate is further decreased to around 0.0457 d$^{-1}$ to achieve the desired reference of 1.5 gCOD/L. (iv) Finally, at $t$ = 1500 d, when the disturbance in the substrate input occurs, dilution is also decreased because high HRT is required to degrade the increase of the input substrate. It is also noted that the methanogenic biomass concentration significantly increases due to the additional substrate input.

### 3.2. Control of the Methanogenic Biomass Concentration

3.2.1. Control Problem

In this case, the control problem is set as the regulation of the methanogenic biomass concentration (which is proportional to the methane production) to the desired reference by manipulating the dilution rate. It is noted that besides the energy recovery from microalgae biomass after lipid extraction in a bio-refinery concept, when the cell lipid content does not exceed 40%, AD of the whole biomass is an attractive alternative for the energetic recovery of cell biomass via biogas generation [12,16,25].

3.2.2. Numerical Results

For this control problem, the first-order transfer function parameters are $k_p$ = −22.72 and $\tau_0$ = 75, which are computed based on the input–output response shown in Figure 1. Figure 4 shows the closed-loop performance of the MEC controller for three sets of controller parameters, as well as the comparison against a conventional PI controller tuned with IMC rules.

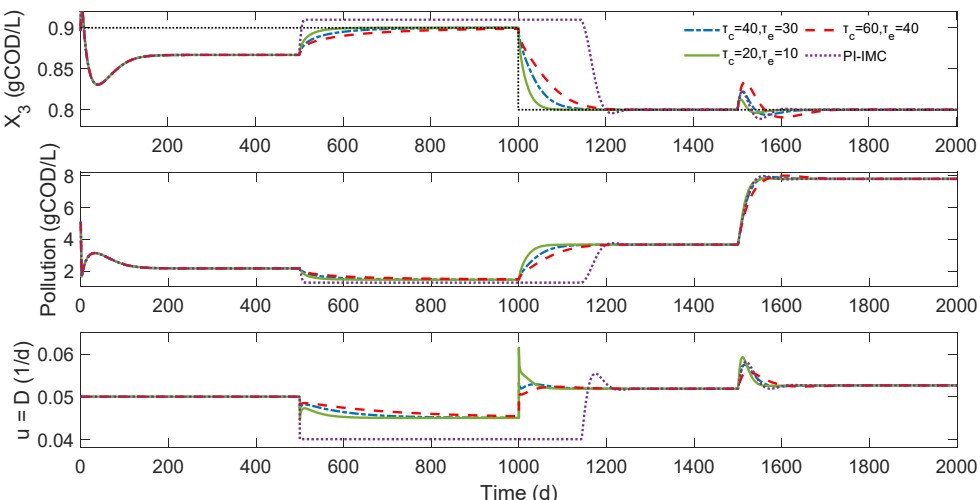

**Figure 4.** Closed-loop performance of the MEC controller for the methanogenic biomass concentration control problem.

As in the organic pollution level control problem, the MEC controller allows the regulation of the methanogenic biomass concentration to the desired references and the rejection of the external perturbation. Once the controller is activated at $t$ = 500 d, the sequence of control inputs is as follows: (i) The first desired reference of 0.9 gCOD/L is achieved with a decrease of the nominal value of the dilution rate from 0.05 to 0.045 d$^{-1}$, to allow the increase of the methanogenic biomass concentration from the nominal value of 0.866 gCOD/L to 0.9 gCOD/L. (ii) The second set-point associated with a lower methanogenic biomass concentration is achieved with an increase in the dilution rate to 0.0518 d$^{-1}$. (iii) Finally,

when the disturbance in the substrate input occurs, dilution is slightly increased because less HRT is required to maintain the same desired microalgae biomass concentration.

Regarding the comparison of the proposed controller with the classical PI controller it is observed the following in both control problems: (i) When the controller is activated, the computed control input with the PI control is more aggressive than the MEC controller such that the minimum dilution value is achieved, which degrades the closed-loop behavior. (ii) The PI controller shows acceptable behavior for rejecting the external perturbation in the substrate input.

## 4. Discussion

Based on the numerical simulations of the proposed control scheme, the following comments are in order:

- Dilution and $S_{in}$ values: The AD operation is markedly influenced by the dilution rate and the substrate feed [19,20]. The observed values in the base and controlled process numerical simulation correspond to the region of lower organic pollution level and higher methanogenic biomass concentration according to the in-depth study presented by Khedim et al. [30] for the selected parameter values. As the substrate input feed increases and the dilution rate decreases, an increase in the methanogenic biomass concentration can achieve, as is shown in Figure 3 when the external perturbation is applied. Khedim et al. [30] suggest that the optimum yield of the MAD model in terms of biogas production was obtained for the following ranges [0.001–0.05] d$^{-1}$, [0.03–30] gCOD/L of $D$, and $S_{in}$, respectively. Low dilution rates correspond to high HRT, allowing the active biomass population to remain in the reactor and not limiting the hydrolysis step.
- MEC closed-loop performance: The numerical results of the proposed controller on the benchmark MAD model demonstrate the capabilities and versatility of the MEC control approach when controlling the complex operation of AD. It is also noted that only two papers have addressed control designs for the AD of microalgae for the organic pollution level control problem [22,43]. In both cases, considering a possible error in that references in the time units (from hours to days) and sight differences between the values of some variables, the magnitude of the computed dilution rate is similar to the numerical assessment of two proposed nonlinear controllers based on feedback linearization and robust adaptive controllers. However, since both contributions include state and uncertain kinetic estimators, a fair comparison is not possible.

## 5. Conclusions

This paper addressed two control problems in the anaerobic digestion of microalgae: (i) the regulation of the organic pollution level and (ii) the regulation of the methanogenic biomass concentration. In the first case, the control problem is aimed at the final disposal of the residual microalgae biomass by reducing its organic components. In the second case, the control problem seeks to enhance energy recovery from residual biomass or microalgae with low lipid content. Aside from its efficiency and good robustness properties, the proposed controller is also characterized by simplicity, being thus appropriate for implementation in real-life systems. Another significant advantage is its generality. This technique may be applied to similar and more complex anaerobic digestion processes where a low-order input–output model can be obtained. Although the proposed controller was evaluated on a benchmark validated model of the microalgae anaerobic digestion, the effect of relevant operation variables, such as the accumulation of VFAs, must be considered. Future research can focus on applying the proposed robust controller to real and possible multiple input–multiple output scenarios.

**Author Contributions:** Conceptualization, A.V.-P. and H.P.; methodology, M.R.-J. and H.P.; validation, A.V.-P., S.E.V.-C., and J.V.; investigation, M.R.-J. and H.P.; writing—original draft preparation,

H.P.; writing—review and editing, M.R.-J., A.V.-P., S.E.V.-C., and J.V. All authors have read and agreed to the published version of the manuscript.

**Funding:** This research received no external funding.

**Acknowledgments:** Mariana Rodriguez-Jara acknowledges financial support from Consejo Nacional de Ciencia y Tecnología (CONACyT) for supporting her Ph.D. studies in process engineering at the UAM-Azcapotzalco.

**Conflicts of Interest:** The authors declare that they have no known competing financial interests or personal relationships that could appear to have influenced the work reported in this paper.

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
