# Peer review of "Robust Control Based on Modeling Error Compensation of Microalgae Anaerobic Digestion"

_fermentation, doi:10.3390/fermentation9010034_

Round 1

Reviewer 1 Report

The title “Robust control of microalgae anaerobic digestion” is rather confusing. Because of the characteristic of a research article, the author should specify the key topic in the title. For example, what is the factor of the “robust control”? What would be the consequences of the “robust control”? What was found by the authors?

‘Simple and robust’ control is too repetitive in the context even in the Abstract.

Line 19-20: What were the ‘two problems?’ Throughout the manuscript, they probably are (1) organic pollution level control and (2) methanogenic biomass concentration control, thus it would be required to address them in the Abstract.

A sentence in Line 36-38 is not making any sense thus it needs to be revised.

Line 34: anaerobic digestion (AD) -> abbreviation information should be addressed first.

In the Introduction section, the challenges and limitations of combining AD and microalgae cultivation would have addressed in one paragraph. The description on the objective of ‘control designs’ is rather vague.

Maybe typo in Line 65: scared -> scarcely

From the perspective of AD, is it proper to represent COD concentration as ‘organic pollution level’, not as total substrate concentration?

Section 2.1 description focuses on AD process occurred by bacterial culture, not on the mechanism of microalgae in the process.

Line 127: It does not make sense to italicize ‘pH’.

The authors employed Chlorella vulgaris as a model strain. Would it be competent and dominant species in the mixed culture in the AD process? Information on why the authors selected the strain and growth characteristics (as well as pH) within the common AD environment would be required. A description of the stability and versatility of the strain could be helpful. Is there any possibility of having a positive or negative effect by a specific strain?

Comparison with the factors among the bacteria would be effective to present the impact of methanogenic bacteria on AD. In other words, is there any result indicating that fermentative and acetogenic bacteria have no impact on AD?

Reviewer 2 Report

The authors applied a simple and robust control design to the anaerobic digestion of microalgae, based on the modeling error compensation technique in order to have simple and robust controllers.

The paper is well structured and the discussion is appropriate.

I suggest just to cite the recent review about anaerobic digestion of algae "Sargassum invasion in the Caribbean: an opportunity for coastal communities to produce bioenergy based on biorefinery - an overview" (2022). Moreover, in line 33 it is not correct write about transerification into microalgae lipid, because transesterification converts lipid (triacylglicerols) in fatty acid esters and glycerol.

Reviewer 3 Report

I have reviewed this manuscript with great pleasure as it is very carefully and clearly written. It is very straight-forward to follow. It is properly set into context (introduction, conclusions). The figures are very nicely done. I recommend the publication of the paper as it is.

Round 2

Reviewer 1 Report

The manuscript has been well-revised according to the previous comments. Before acceptance, some minor parts should be considered by authors.

In the Abstract, providing quantitative information for the “improving” could be helpful for the readers to understand the study.

Line 62: correct “aminoacid” to “amino acid”

Line 142-144: Changing the order of the two sentences would make the flow more convincing.

Line 143, 152: “Chlorella vulgaris” to “C. vulgaris”
